# Associations of presenting symptoms and subsequent adverse clinical outcomes in people with unipolar depression: a prospective natural language processing (NLP), transdiagnostic, network analysis of electronic health record (EHR) data

Rashmi Patel [1,2] Jessica Irving [1] Aimee Brinn,[1] Matthew Taylor,[3] Hitesh Shetty,[2] Megan Pritchard,[2,4] Robert Stewart,[2,4] Paolo Fusar-Poli,[1,2] Philip McGuire[1,2]

RP, JI and AB contributed equally.

For numbered affiliations see end of article.

**Correspondence to**
Dr Rashmi Patel;
bmj@rpatel.co.uk

## ABSTRACT

**Objective** To investigate the associations of symptoms of mania and depression with clinical outcomes in people with unipolar depression.

**Design** A natural language processing electronic health record study. We used network analysis to determine symptom network structure and multivariable Cox regression to investigate associations with clinical outcomes.

**Setting** The South London and Maudsley Clinical Record Interactive Search database.

**Participants** All patients presenting with unipolar depression between 1 April 2006 and 31 March 2018.

**Exposure** (1) Symptoms of mania: Elation; Grandiosity; Flight of ideas; Irritability; Pressured speech. (2) Symptoms of depression: Disturbed mood; Anhedonia; Guilt; Hopelessness; Helplessness; Worthlessness; Tearfulness; Low energy; Reduced appetite; Weight loss. (3) Symptoms of mania or depression (overlapping symptoms): Poor concentration; Insomnia; Disturbed sleep; Agitation; Mood instability.

**Main outcomes** (1) Bipolar or psychotic disorder diagnosis. (2) Psychiatric hospital admission.

**Results** Out of 19 707 patients, at least 1 depression, overlapping or mania symptom was present in 18 998 (96.4%), 15 954 (81.0%) and 4671 (23.7%) patients, respectively. 2772 (14.1%) patients subsequently developed bipolar or psychotic disorder during the follow-up period. The presence of at least one mania (HR 2.00, 95% CI 1.85 to 2.16), overlapping symptom (HR 1.71, 95% CI 1.52 to 1.92) or symptom of depression (HR 1.31, 95% CI 1.07 to 1.61) were associated with significantly increased risk of onset of a bipolar or psychotic disorder. Mania (HR 1.95, 95% CI 1.77 to 2.15) and overlapping symptoms (HR 1.76, 95% CI 1.52 to 2.04) were associated with greater risk for psychiatric hospital admission than symptoms of depression (HR 1.41, 95% CI 1.06 to 1.88).

### Strengths and limitations of this study

► By applying natural language processing (NLP) techniques to electronic health records, we were able to obtain symptom data routinely collected by clinicians for a large sample of 19 707 individuals, permitting investigation into associations between specific symptom domains and clinical outcomes.

► NLP enabled us to detect subgroups of patients with unipolar depression also experiencing symptoms of mania (ie, depression with mixed features) that cannot be discretely identified through ICD-10 diagnostic classification.

► Network analysis allowed us to identify the degree to which certain symptoms co-occur and influence each other as part of the clinical presentation.

► NLP techniques are unable to achieve 100% precision and recall and so the presence of symptoms in electronic health records may not perfectly reflect the true prevalence of symptoms within the cohort.

► Our outcome data derive from a secondary mental healthcare setting and cannot be generalised to people with unipolar depression that are managed within primary care or in the community.

**Conclusions** The presence of mania or overlapping symptoms in people with unipolar depression is associated with worse clinical outcomes. Symptom-based approaches to defining clinical phenotype may facilitate a more personalised treatment approach and better predict subsequent clinical outcomes than psychiatric diagnosis alone.

## INTRODUCTION

Unipolar depression is one of the most prevalent mental disorders and can lead to

impaired functioning and considerable illness burden.[1] A proportion of people with unipolar depression present with concurrent subthreshold manic/hypomanic symptoms known as 'mixed features'. These can include elevated mood, grandiosity, pressure of speech, flight of ideas, increased energy, increased risk taking and decreased need for sleep. The estimated prevalence of at least one manic symptom and at least three manic systems among people with unipolar depression is 22%–50% and 7%–23%, respectively.[2] Evidence indicates that unipolar depression with mixed features requires only two or three concomitant manic symptoms to be associated with a more severe and disabling course of illness,[3 4] increased relapse frequency,[3 5] increased risk for suicidality,[5] poorer antidepressant response[6 7] and greater functional difficulties.[4 5] More recently, attenuated manic and cyclothymic features in young people with unipolar depression have been operationalised by the semistructured interview for bipolar at-risk states. This psychometric interview defines subgroup of clinical high-risk states for developing a first episode of bipolar disorder; pilot findings suggest a promising prognostic accuracy.[8] The clustering of symptoms indicated by the Semistructured Interview for Bipolar At Risk States (SIBARS) suggests that aggregation of specific symptoms may be useful to detect early phases preceding the onset of bipolar disorder and other severe clinical outcomes. This could theoretically allow early identification of those more likely to develop bipolar disorders and therefore implement preventive approaches that could benefit their outcomes.

Investigating the earlier phases preceding the onset of a bipolar disorder is complicated by the need of prospective designs that represent real world clinical course of this disorder. Electronic health records (EHRs) offer empirical advantages tackling these issues by allowing large scale investigation of pattern of symptoms that may preceded the onset of the disorder. Natural language processing (NLP) is an innovative, automated information extraction technique to investigate the associations of specific symptoms of mental disorders with clinical outcomes using EHR data from mental healthcare.[9–12]

These associations could be further elucidated by adopting network analyses, which investigate the co-occurrence and cross-influence of individual symptoms as part of the clinical presentation. Network analysis has previously demonstrated that the presence of mixed features in people with unipolar depression is associated with poorer response to lurasidone compared with people without mixed features (n=208).[13] A study applying network analysis techniques to mood symptom data obtained through online questionnaires (n=647) and outpatient clinical assessments (n=1370) demonstrated that symptoms of mania and depression are transdiagnostic and can occur in either unipolar depression or bipolar disorder.[14] However, to the best of our knowledge, there have been no studies which have investigated the relationship between symptoms of mania and depression

through network analysis applied to large volumes of EHR data.

We aimed to identify how often symptoms of mania are recorded in the EHR of patients who initially present with unipolar depression, and the association of symptoms of mania with subsequent diagnosis of bipolar or psychotic disorder, and with psychiatric hospital admissions. We represent symptoms of depression and mania graphically using network analysis in a sample of 19 707 patients presenting to a secondary mental healthcare service in South London.

## METHODS

### Participants and setting

We extracted data from deidentified EHRs for individuals accessing secondary mental healthcare provided by the South London and Maudsley NHS Foundation Trust (SLaM; London, UK) between 1 April 2006 and 31 March 2018. SLaM holds EHRs for over 500 000 people in receipt of mental healthcare since 2007.[15] The Trust provides community and inpatient mental healthcare within its catchment area covering the London boroughs of Lambeth, Southwark, Croydon and Lewisham. We used the Clinical Record Interactive Search tool (CRIS) to extract and analyse deidentified EHR data from SLaM, with full access to the database population used to create the study population.[15]

Inclusion criterion: all individuals with a primary diagnosis of unipolar depression (online supplemental eTable 1) recorded in a structured diagnosis EHR field who presented to a SLaM community or inpatient service between 1 April 2006 and 31 March 2018. Follow-up data were obtained from the date of acceptance to the community or inpatient service in which the first recorded diagnosis of unipolar depression was made (the index date).

Exclusion criteria: patients with a prior International Classification of Diseases 10th revision (ICD-10) diagnosis of bipolar disorder (F31*), mania (F30*) or psychotic disorder (F1x.5, F2x, F32.3 and F33.3) were excluded. We further excluded patients who went on to receive a diagnosis of psychotic disorder, bipolar disorder or mania within 3 months following their first depression diagnosis. This exclusion criterion was employed to exclude individuals whose first presentation diagnosed as unipolar depression could have represented the initial phase of a bipolar disorder. We excluded patients with a delay of 1 year or more between the index date and first unipolar depression diagnosis date as the symptoms recorded around the time of presentation may not have represented symptoms associated with the recorded diagnosis of unipolar depression for individuals with a long delay between initial referral and subsequent unipolar depression diagnosis. Patients with no recorded symptoms or missing ethnicity data were also excluded.

## Measures

Data were extracted on age at index date, gender, ethnicity, symptoms of depression or mania (including those which could be a feature of either depression or mania), psychiatric hospital admissions (voluntary and involuntary under the UK Mental Health Act (MHA)) and diagnoses of bipolar disorder (ICD-10: F30/31) or psychotic disorders (ICD-10: F1x.5, F2x, F32.3 and F33.3). Ethnicity was recoded according to categories defined by the UK Office of National Statistics as Asian, black, mixed, other and white (online supplemental eTable 2).

We obtained data for 20 symptoms using NLP tools developed for use on EHR data obtained using CRIS (online supplemental eTable 3). These symptoms were chosen based on their inclusion in either ICD-10 or Diagnostic and Statistical Manual of Mental Disorders, Fifth Edition (DSM-5) diagnostic criteria for unipolar depression and/or mania with the exception of mood instability which is not present in diagnostic criteria for either disorder but was included because it has been previously shown to be an important clinical feature in people with unipolar depression and bipolar disorder.[9] The CRIS NLP tools extract symptom data from unstructured free text clinical documents which record clinical assessment, mental state examination and the agreed treatment plan. Their performance is measured through precision and recall. In this case, precision refers to the number of true positive instances that the NLP application has identified, divided by the total number of instances retrieved by the NLP application (both true positive and false positive instances). Recall refers to the number of true positive instances identified by the NLP application, divided by the total number of existing true positives in the entire sample. Precision and recall statistics for the NLP applications employed in this as well as further details on their derivation, are provided in the CRIS NLP Applications Library.[16]

Symptom data were drawn from a period within 3 months either side of the index date, as symptoms presenting in this time are more likely to inform initial presentation. We converted each symptom into dichotomous variables of *present* vs *absent* in a given patient's record. This approach was chosen instead of quantifying the number of times each symptom was recorded in order to minimise confounding by the volume of documentation within a patient's record which may vary depending on illness severity and mode of clinical presentation (eg, outpatient vs inpatient). We classified symptoms a priori into three groups according to ICD-10/DSM-5 diagnostic criteria (see online supplemental eTable 3): depressive, mania and overlapping symptoms (ie, symptoms present in ICD-10/DSM-5 classification criteria for both unipolar depression and bipolar disorder).

Follow-up data were obtained from 3 months following the index date for each patient up to 31 March 2021 to ensure predictor symptom data were temporally separate from outcome data. Psychiatric hospitalisation data were obtained as the date of first recorded admission to a SLaM psychiatric hospital during the follow-up period and categorised as (1) any psychiatric hospital admission and (2) compulsory psychiatric hospital admission under section 2 or 3 of the UK MHA. Data on subsequent diagnosis of psychotic disorder or bipolar disorder were obtained from structured diagnostic fields and unstructured free text EHR documents using NLP.[16] We chose to obtain follow-up diagnosis data from both structured and unstructured fields to maximise early identification of onset of these disorders as clinicians typically document emerging diagnoses in unstructured free text prior to recording the diagnosis in the structured EHR fields.

## Statistical analysis

We followed the "TRANSD" iagnostic research recommendations in psychiatry (TRANSD) reporting guidelines (online supplemental eTable 4).[17] All analyses were conducted in R (V.3.6.1). We obtained descriptive statistics on age, gender, ethnicity and the most commonly recorded symptoms and symptom combinations. To investigate demographic associations with recording of manic symptoms among patients diagnosed with unipolar depression, we assessed associations between the presence of at least once manic symptom and at least one overlapping symptom with age, gender and ethnicity through univariate logistic regression.

We used network analysis to visually examine connectivity between depressive and manic symptoms. In a network graph, circles represent nodes (symptom items) and lines represent edges. Each edge denotes a partial correlation between two symptoms (ie, the correlation between two symptoms when controlling for all other symptoms in the network). Edge thickness represents correlation magnitude. We estimated the network using the Enhanced Least Absolute Shrinkage and Selection Operator procedure implemented via the EstimateNetwork package. The qgraph package was used to visualise the network.

Node placement was determined using the Fruchterman-Reingold algorithm, which places the most central nodes visually centre in the network graph.[18] In terms of centrality indices, we calculated strength, closeness and betweenness defined in online supplemental eTable 5. We used a case-dropping bootstrapping procedure to assess network stability. Further information on our network analysis approach can be found in online supplemental eMethods 1.

We created dichotomous predictor variables indicating where individuals had at least one reported symptom within each of the following symptom groups: mania, depression, overlapping. We chose to dichotomise these predictors (ie, the presence of at least one symptom in each group was recorded as '1' and absence of any symptom within a group recorded as '0') as symptom groups had unequal numbers of constituent symptoms, therefore, treating these predictors as continuous would have weighted one symptom group over the other. Additionally, we examined the associations of each individual

symptom within the mania and overlapping group with the outcomes described subsequently.

We included three demographic covariates in multivariable analyses to evaluate associations of symptom groups with clinical outcomes: age, gender and ethnicity. Using these covariates and the predictors described above, we estimated Cox proportional hazards regression models (censor date: 31 March 2021) to predict HRs for risk of the following outcomes:

1. Receiving a diagnosis of bipolar or psychotic disorder.
2. Psychiatric hospital admission (any).
3. Compulsory psychiatric admission under the UK MHA.
4. Receiving a diagnosis of bipolar disorder.
5. Receiving a diagnosis of psychotic disorder.

Multivariable analyses were corrected for multiple-testing using the Bonferroni method.

Bipolar disorder was defined according to the following ICD-10 diagnostic categories: ICD-10 F30 and F31.

Psychotic disorder was defined according to the following ICD-10 diagnostic categories: ICD-10 F1x.5, F2x, F32.3 and F33.3.

Where a patient was diagnosed with both bipolar disorder and psychotic disorder during the follow-up period, the first recorded diagnosis was considered when assessing outcome (1) listed above.

## Patient and public involvement

The study was approved by the SLaM CRIS Oversight Committee which has service user representation. Researchers wishing to access CRIS data can make an application to the Oversight Committee via the CRIS website (https://projects.slam.nhs.uk/research/cris/cris-project-application). The design and completion of approved CRIS projects is supported by a service user advisory group (the CRIS SUCAG) whose members have experience of receiving mental healthcare (or providing care to someone receiving mental healthcare) from SLaM clinical services.[19]

## RESULTS
### Descriptive statistics

The final sample comprised 19 707 patients, having excluded 1206 patients with over a year delay between the index date and unipolar depression diagnosis, 3407 patients with no documented symptom data, 2364 patients with missing ethnicity data and a further 7 patients with missing gender data (see online supplemental figure 1).

Demographic and clinical characteristics are reported in table 1. Our sample was predominantly female (61.9%), of White ethnicity (59.5%), with a mean age (SD) of 38.0 (12.3) years. Recorded symptoms of depression were ascertained in 18 998 patients (96.4%), diagnostically overlapping symptoms in 15 954 patients (81.0%) and symptoms of mania in 4671 patients (23.7%). Patients with at least one symptom of mania were significantly more likely to be male (OR 1.15, 95% CI 1.08 to 1.23, p<0.001) and significantly less likely to be of Other

| Table 1 | Demographic and clinical characteristics |
|---|---|
| **Characteristic** | **Value** |
| Demographics | |
| Age (mean, SD) | 38.0 (12.3) years |
| Female (n, (%)) | 12 196 (61.5) |
| Ethnicity (n, (%)) | |
| Asian | 1133 (5.7) |
| Black | 3473 (17.6) |
| Mixed | 439 (2.2) |
| Other | 2494 (12.6) |
| White | 12 168 (61.7) |
| Depressive symptoms (n (%)) | |
| Disturbed mood | 17 188 (87.2) |
| Tearfulness | 10 196 (51.7) |
| Reduced appetite | 7996 (40.6) |
| Hopelessness | 6160 (31.3) |
| Low energy | 6159 (31.3) |
| Guilt | 5828 (29.6) |
| Weight loss | 3778 (19.2) |
| Anhedonia | 3514 (17.8) |
| Worthlessness | 2560 (13) |
| Helplessness | 2168 (11) |
| Diagnostically overlapping symptoms* (n (%)) | |
| Disturbed sleep | 12 271 (62.3) |
| Poor concentration | 8056 (40.9) |
| Mood instability | 5315 (27.0) |
| Agitation | 5074 (25.7) |
| Insomnia | 3250 (16.5) |
| Manic symptoms (n (%)) | |
| Irritability | 3911 (19.8) |
| Elation | 642 (3.3) |
| Pressured speech | 348 (1.8) |
| Flight of ideas | 356 (1.8) |
| Grandiosity | 261 (1.3) |

*Symptoms that overlap International Classification of Diseases 10th revision (ICD-10)/Diagnostic and Statistical Manual of Mental Disorders, Fifth Edition (DSM-5) criteria for both unipolar and bipolar depression or mania.

ethnicity relative to white ethnicity (OR 0.85, 95% CI 0.76 to 0.94, p=0.006). Patients with at least one diagnostically overlapping symptom were also significantly more likely to be male (OR 1.20, 95% CI 1.11 to 1.29, p<0.001), significantly more likely to be of Asian ethnicity relative to white ethnicity (OR: 1.26, 95% CI 1.07 to 1.49, p=0.019).

We extracted a median of 11 (IQR: 5–24) symptoms per patient, reported in 6910 different combinations. There were 642 combinations of symptoms of depression, 32 combinations of overlapping symptoms and 31 combinations of symptoms of mania. We found at least

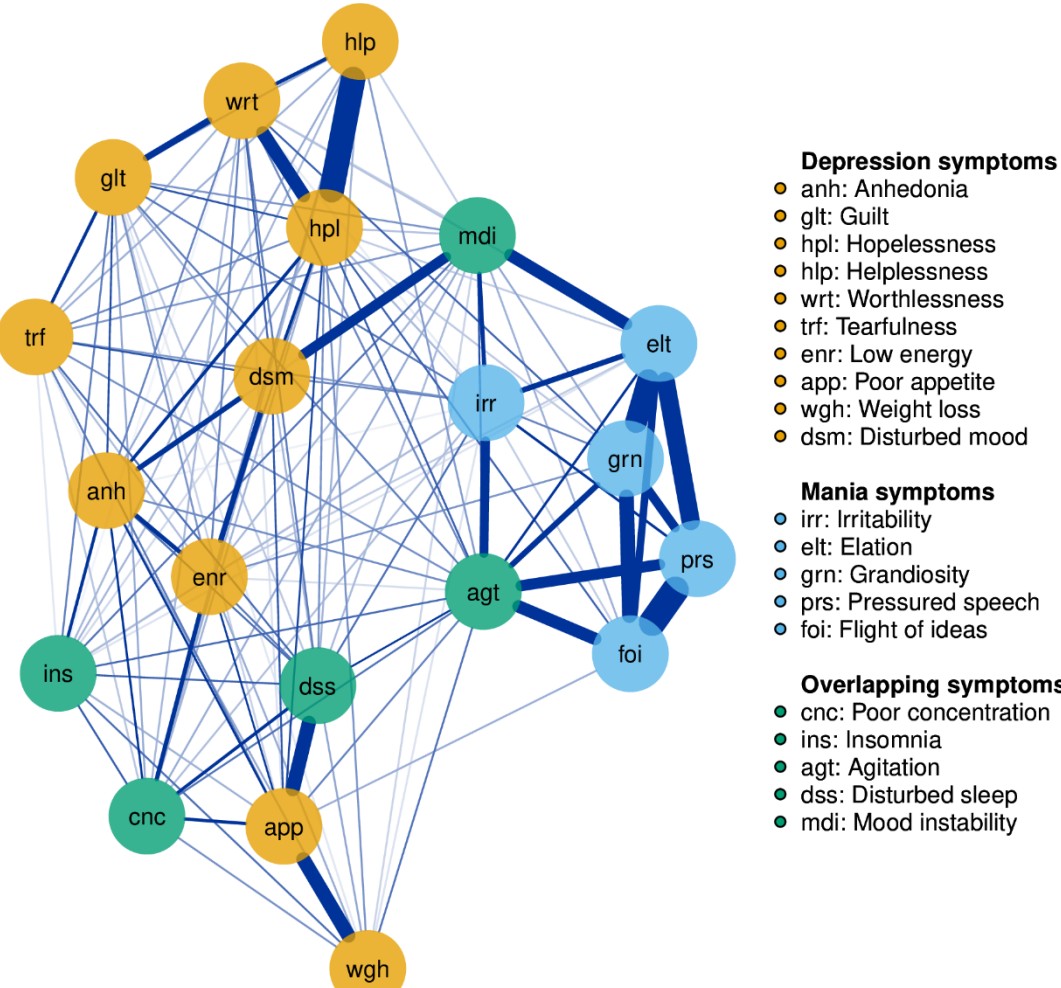

**Depression symptoms**
- anh: Anhedonia
- glt: Guilt
- hpl: Hopelessness
- hlp: Helplessness
- wrt: Worthlessness
- trf: Tearfulness
- enr: Low energy
- app: Poor appetite
- wgh: Weight loss
- dsm: Disturbed mood

**Mania symptoms**
- irr: Irritability
- elt: Elation
- grn: Grandiosity
- prs: Pressured speech
- foi: Flight of ideas

**Overlapping symptoms**
- cnc: Poor concentration
- ins: Insomnia
- agt: Agitation
- dss: Disturbed sleep
- mdi: Mood instability

**Figure 1** Circles represent nodes (symptom items) and lines represent edges. Each edge denotes a partial correlation between two symptoms (ie, the correlation between two symptoms when controlling for all other symptoms in the network); edge thickness represents correlation magnitude. We used the Fruchterman-Reingold algorithm, which places the most central nodes visually centre in the network graph, to determine node placement.[18]

one overlapping symptom recorded for 15 954 patients (81.0%). At least two overlapping symptoms were recorded for 10 119 patients (51.3%) and at least three were recorded for 5188 patients (26.3%). At least one mania symptom was recorded for 4671 (23.7%) patients. A total of 634 (3.2%) patients had at least two recorded symptoms of mania, while 3 or more symptoms were recorded for 144 (0.73%) patients.

The distribution of ICD-10 unipolar depression diagnoses across the cohort is reported in online supplemental eTable 6. The most common index diagnosis was F32.1 Moderate depressive episode (n=5242, 26.6%).

### Network analysis
Figure 1 presents the regularised partial correlation network of symptoms in people with unipolar depression. Symptoms of mania cluster between themselves, and symptoms of depression and some diagnostically overlapping symptoms cluster together (including within-group clustering). Visual inspection of the network graph revealed two diagnostically overlapping symptoms (agitation and

mood instability) with strong associations to the principal symptoms of mania.

The resulting network remained stable after dropping up to 33% of cases from the sample. The strongest edges for each symptom are presented in table 2. The strongest association was between pressured speech and flight of ideas; the second strongest association was between hopelessness and helplessness. Symptom centrality indices (strength, closeness, betweenness and expected influence) are provided in online supplemental figure 2. Agitation was the most central symptom in terms of strength, betweenness and expected influence. Disturbed mood was the most central symptom in terms of closeness. Stability analysis of the centrality indices suggests correlation stability coefficients of 0.67 for strength, 0.67 for betweenness and 0.67 for closeness.[20]

### Diagnostic outcomes
During 166 686.5 person-years of follow-up (mean: 8.45 years, SD: 15.0 years), 2772 (14.07%) individuals received a subsequent diagnosis of a psychotic or bipolar disorder;

**Table 2** Strongest associations for each symptom in decreasing order of strength

| Node 1 | Node 2 | Mean edge weight (95% CI) |
|---|---|---|
| Pressured speech | Flight of ideas | 1.76 (1.35 to 2.15) |
| Hopelessness | Helplessness | 1.65 (1.52 to 1.75) |
| Elation | Grandiosity | 1.50 (1.11 to 1.87) |
| Poor appetite | Weight loss | 1.11 (0.99 to 1.17) |
| Grandiosity | Flight of ideas | 1.06 (0.53 to 1.68) |
| Disturbed sleep | Poor appetite | 1.08 (1.00 to 1.15) |
| Flight of ideas | Agitation | 0.97 (0.72 to 1.26) |
| Irritability | Agitation | 0.80 (0.72 to 0.89) |
| Mood instability | Disturbed mood | 0.86 (0.74 to 1.01) |
| Guilt | Worthlessness | 0.67 (0.58 to 0.77) |
| Low energy | Disturbed mood | 0.62 (0.49 to 0.75) |
| Anhedonia | Disturbed mood | 0.61 (0.41 to 0.80) |
| Concentration | Energy | 0.57 (0.49 to 0.63) |
| Insomnia | Anhedonia | 0.50 (0.39 to 0.60) |
| Helplessness | Worthlessness | 0.53 (0.41 to 0.65) |
| Agitation | Disturbed sleep | 0.43 (0.35 to 0.52) |
| Tearfulness | Disturbed mood | 0.37 (0.26 to 0.46) |
| Worthlessness | Disturbed mood | 0.31 (0.12 to 0.54) |
| Weightloss | Disturbed mood | 0.02 (-0.09 to 0.09) |

2150 (10.91%) received a psychotic disorder diagnosis and 986 (5.00%) received a bipolar disorder diagnosis. The incidence rate (cases per 10 000 person-years risk) was 166.4 (95% CI 160.2 to 172.7) for diagnosis of psychotic or bipolar disorder, 128.6 (95% CI 123.2 to 134.2) for diagnosis of psychotic disorder and 59.0 (95% CI 55.4 to 62.8) for diagnosis of bipolar disorder.

Results of multivariable Cox proportional hazard analyses, corrected for multiple-testing, are reported in table 3. In decreasing order of risk, individuals with at least one symptom of mania, at least one overlapping symptom, or at least one symptom of depression had significantly increased risk for bipolar or psychotic disorder. Pressured speech (HR 3.07, 95% CI 2.56 to 3.69), grandiosity (HR 3.04, 95% CI 2.46 to 3.75) and flight of ideas (HR 2.83, 95% CI 2.30 to 3.48) were the individual symptoms most strongly associated with subsequent onset of psychotic or bipolar disorder. Black or Asian ethnicity (relative to white ethnicity) and age were also significant predictors of risk of subsequent diagnosis of bipolar or psychotic disorder.

Cox proportion hazard analyses for risk of subsequent diagnosis of bipolar disorder are reported in online supplemental eTable 7. In a multivariable analysis, individuals with at least one symptom of mania or at least one overlapping symptom had a significantly increased risk of subsequent bipolar disorder diagnosis. Individual symptoms: grandiosity (HR 4.36, 95% CI 3.24 to 5.87), elation (HR 4.21, 95% CI 3.46 to 5.12) and pressured speech (HR 4.07, 95% CI 3.12 to

5.31) were most strongly associated with subsequent onset of bipolar disorder.

Cox proportion hazard analyses for risk of subsequent diagnosis of psychotic disorder are reported in online supplemental eTable 8. In decreasing order of risk, individuals with at least one mania symptom, at least one overlapping symptom or at least one depression symptom had a significantly increased risk of onset of psychotic disorder. All individual overlapping/mania symptoms were significantly associated with subsequent psychotic disorder diagnosis. Of these individual symptoms, pressured speech (HR 2.69, 95% CI 2.17 to 3.33), flight of ideas (HR 2.61, 95% CI 2.07 to 3.30) and grandiosity (HR 2.61, 95% CI 2.03 to 3.36), were most strongly associated with the onset of psychotic disorder.

### Psychiatric hospital admission

There were 1719 psychiatric admissions, of which 516 were compulsory under the UK MHA. Table 4 presents the results from the Cox regression analyses for risk of psychiatric hospital admission and online supplemental eTable 9 for risk of compulsory psychiatric hospital admission under the UK MHA. Symptoms of depression, symptoms of mania and overlapping symptoms were all associated with non-compulsory hospitalisation risk. Mania and overlapping symptoms predicted compulsory hospitalisation, while depressive symptoms did not. Of the individual overlapping/mania symptoms, pressured speech (HR 3.02, 95% CI 2.41 to 3.77), agitation (HR 2.91, 95% CI 2.65 to 3.20) and flight of ideas (HR 2.54, 95% CI 1.98 to 3.25) were associated with the greatest risk of any psychiatric hospital admission.

Pressured speech (HR 4.37, 95% CI 3.12 to 6.14), elation (HR 3.96, 95% CI 2.98 to 5.25) and flight of ideas (HR 3.80, 95% CI 2.61 to 5.53) were associated with the greatest risk of compulsory psychiatric hospital admission. Black and Asian ethnicity were associated with increased risk of compulsory psychiatric hospital admission, compared with white ethnicity. Male gender was associated with increased risk of compulsory psychiatric hospital admission, compared with female gender.

### DISCUSSION

We found that people with unipolar depression have a heterogeneous clinical phenotype with a significant number (2772 during 166 686.5 person-years of follow-up) going on to develop a bipolar or psychotic disorder. In a large sample of 19 707 individuals with a diagnosis of unipolar depression, 23.7% had at least one reported symptom of mania (irritability, elation, grandiosity, pressured speech or flight of ideas), and this subset of patients was most likely to be White and male. Irritability was the most commonly documented symptom of mania.

### Associations of symptoms with psychotic or bipolar disorder diagnosis

Overall, those with symptoms of mania (ie, mixed features) had the greatest rates of subsequent diagnosis of

**Table 3** Multivariable Cox regression models examining factors associated with subsequent psychotic or bipolar disorder diagnosis

| Predictor | Risk of developing psychotic or bipolar disorder (no of events=2772) | | | |
| --- | --- | --- | --- | --- |
| | Univariate HR (95% CI) | P value | Multivariable HR (95% CI)† | P value |
| Demographics | | | | |
| Age | 1.00 (1.00 to 1.01) | 0.209 | 1.00 (1.00 to 1.01) | 0.025 |
| Male gender | 1.01 (0.93 to 1.09) | 0.843 | 1.03 (0.95 to 1.11) | 0.489 |
| Ethnicity | | | | |
| White | Ref | Ref | Ref | Ref |
| Asian | 1.42 (1.22 to 1.64) | <0.001*** | 1.44 (1.24 to 1.67) | <0.001*** |
| Black | 1.52 (1.39 to 1.67) | <0.001*** | 1.55 (1.41 to 1.70) | <0.001*** |
| Mixed | 1.07 (0.83 to 1.39) | 0.644 | 1.10 (0.85 to 1.43) | 0.495 |
| Other | 1.03 (0.90 to 1.17) | 0.692 | 1.04 (0.91 to 1.18) | 0.561 |
| Symptom groups | | | | |
| Mania (≥1 symptom) | 1.98 (1.84 to 2.15) | <0.001*** | 2.00 (1.85 to 2.16) | <0.001*** |
| Overlapping (≥1 symptom) | 1.73 (1.54 to 1.94) | <0.001*** | 1.71 (1.52 to 1.92) | <0.001*** |
| Depression (≥1 symptom) | 1.31 (1.07 to 1.60) | 0.012* | 1.31 (1.07 to 1.61) | 0.010* |
| Individual symptoms | | | | |
| Irritability | 1.85 (1.70 to 2.00) | <0.001*** | 1.85 (1.71 to 2.01) | <0.001*** |
| Elation | 2.55 (2.21 to 2.95) | <0.001*** | 2.65 (2.29 to 3.07) | <0.001*** |
| Pressured speech | 3.05 (2.54 to 3.66) | <0.001*** | 3.07 (2.56 to 3.69) | <0.001*** |
| Flight of ideas | 2.83 (2.31 to 3.48) | <0.001*** | 2.83 (2.30 to 3.48) | <0.001*** |
| Grandiosity | 2.99 (2.42 to 3.69) | <0.001*** | 3.04 (2.46 to 3.75) | <0.001*** |
| Disturbed sleep | 1.52 (1.40 to 1.65) | <0.001*** | 1.49 (1.37 to 1.62) | <0.001*** |
| Poor concentration | 1.50 (1.39 to 1.61) | <0.001*** | 1.48 (1.38 to 1.60) | <0.001*** |
| Mood instability | 1.74 (1.61 to 1.88) | <0.001*** | 1.79 (1.66 to 1.93) | <0.001*** |
| Agitation | 2.27 (2.11 to 2.45) | <0.001*** | 2.29 (2.12 to 2.47) | <0.001*** |
| Insomnia | 1.47 (1.35 to 1.61) | <0.001*** | 1.48 (1.35 to 1.62) | <0.001*** |

*p<0.05, **p<0.01, ***p<0.001.
†Multivariable analysis adjusted for age, gender and ethnicity.

a psychotic or bipolar disorder. Symptoms of mania were associated with double the risk of psychotic or bipolar disorder. These results are consistent with previous studies where mixed features or subsyndromal hypomanic symptoms during a depressive episode have been associated with subsequent bipolar disorder diagnosis.[21–25] The current study replicates these previous findings in the largest sample size to date.

### Association of symptoms with subsequent hospital admissions

Patients with symptoms of mania were admitted to hospital at the greatest rate, compared with patients with overlapping or depression symptoms. Dudek *et al* found that patients whose diagnosis evolved from unipolar depression to bipolar disorder experienced more hospitalisations than those who remained with a diagnosis of unipolar depression.[23] Moreover, evidence has suggested that highly recurrent depression (defined as more than three episodes) shares some clinical characteristics that are

nearer to bipolar disorder,[26] such as hypomanic switches while on antidepressant drugs[27] and presence of a family history of bipolar disorder.[28] Those with highly recurrent depression may be expected to be admitted to hospital at greater rates, which may account for the association between mixed features and hospitalisation observed in this study. Overlapping symptoms and depression symptoms were associated with increased risk of subsequent psychiatric hospital admissions. However, symptoms of depression were not associated with increased risk of compulsory psychiatric hospital admission.

### The relationship between symptoms of mania and depression

While we investigated a total of five symptoms shared by ICD-10 and DSM-5 criteria for unipolar depression and mania, our network analysis revealed that only agitation and mood instability co-occurred with mania symptoms of irritability, elation, pressured speech, flight of ideas and grandiosity. Symptoms of insomnia, disturbed sleep and poor concentration that are shared between unipolar

**Table 4** Multivariable Cox proportional hazards regression examining factors associated with psychiatric hospital admission

| Predictor | Risk of psychiatric hospital admission (no of events=1719) | | | |
| --- | --- | --- | --- | --- |
| | Univariate HR (95% CI) | P value | Multivariable HR (95% CI)† | P value |
| Demographics | | | | |
| Age | 1.00 (1.00 to 1.00) | 0.666 | 1.00 (1.00 to 1.00) | 0.902 |
| Male gender | 1.28 (1.17 to 1.41) | <0.001*** | 1.32 (1.20 to 1.46) | <0.001*** |
| Ethnicity | | | | |
| White | Ref | | Ref | Ref |
| Asian | 0.98 (0.80 to 1.21) | 0.884 | 0.99 (0.80 to 1.22) | 0.902 |
| Black | 1.25 (1.11 to 1.40) | <0.001*** | 1.30 (1.15 to 1.46) | <0.001*** |
| Mixed | 1.02 (0.74 to 1.41) | 0.884 | 1.06 (0.77 to 1.46) | 0.769 |
| Other | 0.58 (0.48 to 0.69) | <0.001*** | 0.63 (0.52 to 0.75) | <0.001*** |
| Symptom groups | | | | |
| Mania (≥1 symptom) | 1.97 (1.79 to 2.18) | <0.001*** | 1.95 (1.77 to 2.15) | <0.001*** |
| Overlapping (≥1 symptom) | 1.79 (1.55 to 2.07) | <0.001*** | 1.76 (1.52 to 2.04) | <0.001*** |
| Depression (≥1 symptom) | 1.38 (1.03 to 1.83) | 0.033* | 1.41 (1.06 to 1.88) | 0.021* |
| Individual symptoms | | | | |
| Irritability | 1.92 (1.73 to 2.13) | <0.001*** | 1.90 (1.71 to 2.10) | <0.001*** |
| Elation | 2.51 (2.09 to 3.02) | <0.001*** | 2.56 (2.13 to 3.07) | <0.001*** |
| Pressured speech | 3.01 (2.40 to 3.76) | <0.001*** | 3.02 (2.41 to 3.77) | <0.001*** |
| Flight of ideas | 2.61 (2.04 to 3.35) | <0.001*** | 2.54 (1.98 to 3.25) | <0.001*** |
| Grandiosity | 2.73 (2.10 to 3.57) | <0.001*** | 2.58 (1.98 to 3.36) | <0.001*** |
| Disturbed sleep | 1.72 (1.55 to 1.92) | <0.001*** | 1.69 (1.52 to 1.88) | <0.001*** |
| Poor concentration | 1.55 (1.41 to 1.71) | <0.001*** | 1.54 (1.40 to 1.69) | <0.001*** |
| Mood instability | 1.86 (1.68 to 2.04) | <0.001*** | 1.88 (1.71 to 2.08) | <0.001*** |
| Agitation | 2.99 (2.72 to 3.29) | <0.001*** | 2.91 (2.65 to 3.20) | <0.001*** |
| Insomnia | 1.69 (1.52 to 1.89) | <0.001*** | 1.68 (1.51 to 1.88) | <0.001*** |

*p<0.05, **p<0.01, ***p<0.001.
†Multivariable analysis adjusted for age, gender and ethnicity.

depression and bipolar disorder were not significantly associated with the principal symptoms of mania. This concurs with the findings of a study that examined the joint structure of symptoms of mania and depression in two large samples totalling over 2100 patients[14] which found symptoms such as increased energy, grandiosity and euphoria represented a latent specific positive activation dimension that was more specific to bipolar disorder. We found that our identified overlapping symptoms of agitation and mood instability were associated with subsequent psychotic or bipolar disorder diagnosis.

Highly central symptoms are most likely to sustain the rest of the network. Betweenness centrality is one of the most useful metrics for detecting nodes that connect one region of a network analysis graph to another. Removal of these nodes may, therefore, represent an opportunity to destabilise the network. Agitation and mood instability were the two overlapping symptoms that showed the highest betweenness centrality. In theory, changes in nodes with the highest betweenness centrality should have the greatest effect on other nodes, in this case—symptoms of depression and mania. This finding may suggest that clinicians should be especially vigilant in identifying symptoms of agitation or mood instability in patients with unipolar depression, as they could be associated with of subsequent diagnosis of bipolar disorder and/or increased risk of subsequent psychiatric hospitalisation. However in cross-sectional datasets like this one, betweenness and closeness have been found to show low stability and wide confidence intervals .[29]

DSM-5 criteria for unipolar depression with mixed features (online supplemental eTable 10) require, along with a major depressive episode, the presence of at least three of seven symptoms of mania including elevated mood, grandiosity, pressured speech, flight of ideas, increased energy, risk-taking behaviour and decreased need for sleep. However, this clinical picture is extremely rare in mixed depressive states.[2] Furthermore, the DSM-5 criteria for a diagnosis of unipolar depression with mixed features does not account for psychomotor agitation, irritability and mood instability, which Koukopoulos and Sani describe as 'core features' of this clinical presentation.[2]

This concurs with our network analysis which finds symptoms of agitation and mood instability co-occur more commonly with symptoms of mania than other overlapping symptoms such as disturbed sleep and poor appetite. Therefore, the DSM-5 criteria for unipolar depression with mixed features exclude patients at risk of substantial functional impairment who may need enhanced support and more tailored invention.[2] Certain treatments such as SSRIs could prove inappropriate for these patients by worsening agitation and increasing the risk of onset of mania[30] and hospitalisation.[2]

Demographic factors also contributed to risk for poor outcomes. Patients of Black and Asian ethnicity were almost three times and two times more likely respectively than White patients to be compulsorily admitted to a psychiatric hospital under the UK MHA. Previous studies have shown that black and minority ethnic people are disproportionately likely to be admitted to a psychiatric hospital under the UK MHA.[31]

## Strengths

Through our use of EHRs we were able to identify a sample of 19 707 individuals at a scale that would have been infeasible to curate through direct patient recruitment. Furthermore, while the diagnosis of depressive disorder with mixed features was introduced in the DSM-5 to capture the phenomenon of depressed patients who exhibit concurrent symptoms of mania or hypomania,[3] no analogous diagnosis exists in the ICD-10.

Our approach used automated NLP tools to identify symptoms documented in the EHR allowing us to identify patients for further study. Here, we identified 4888 patients diagnosed with unipolar depression with symptoms of mania reported in their EHRs. We were able to stratify these patients into a separate group and represent the heterogeneity of unipolar depression. As a result, we were able to identify differences in clinical outcomes between patients diagnosed with the same heterogeneous diagnostic construct.

Our approach using EHRs provides sample sizes permissive of more granular analyses into specific symptom domains than in previous studies. The use of NLP allowed us to obtain symptom data that tend to be unrepresented in large scale administrative resources. The symptom NLP algorithms allow more fine-grained definition of item constructs than that of typical depression criteria. Mania symptoms are not routinely elicited in standard depression screening questionnaires, and classification systems sometimes combine opposing poles of disorder into unitary terms (like agitation/psychomotor retardation) when they are likely to reflect different underlying neurobiological processes.[32] Our approach allowed us to extract data on agitation and psychomotor retardation separately.

We followed the TRANSD guidelines for reporting transdiagnostic research in psychiatry.[17] We provided a transparent definition of specific ICD-10 diagnostic codes which were fully detailed and numerated to characterise the cohort of patients with unipolar depression and the outcomes of bipolar or psychotic disorders ('T' and 'N' criteria). We reported the primary outcome and the study design and defined the transdiagnostic construct as symptoms overlapping unipolar depression and bipolar disorder and reported the primary outcome of the study (receiving a subsequent bipolar or psychotic disorder diagnosis) and study design (prospective NLP EHR study) fulfilling the 'R' criterion. We appraised the conceptual framework as across-at least three diagnoses (unipolar depression, bipolar disorders, psychosis) and across two diagnostic spectra (mood disorders and psychotic disorders) fulfilling the 'A' criterion.

The ability to identify subgroups of people with unipolar depression and symptoms of mania (as mixed features) would enable the development and evaluation of interventions specifically designed to treat mixed features such as agitation and mood instability which have a significant impact on outcomes and may not be well treated by traditional antidepressant or psychological therapy. For example, patients with symptoms of elevated mood and agitation were less likely than patients without these symptoms to benefit from lurasidone in terms of their Montgomery–Åsberg Depression Rating Scale (MADRS) and Clinical Global Impression – Severity scale (CGI-S) Scores.[13] Conversely, presence of rapid/pressured speech predicted improvement in both mania and depression symptoms with lurasidone. These opposing effects would have been missed in standard approaches that group patients by diagnostic category, thereby treating all symptoms as equally important indicators of prognosis.

## Limitations

NLP techniques are unable to achieve 100% precision and recall and therefore classify information imperfectly.[33] Therefore, the prevalence of symptoms in this cohort cannot be taken as the true prevalence of symptoms within the cohort. However, the overall symptom network is unlikely to be affected by incorrect NLP classification as the network was stable even when dropping over 6700 cases (33% of our sample). As we were limited by the availability of existing algorithms, we were unable to obtain symptom data available for some depression and mania criteria, for example increased appetite or hypersomnia (depression), or impulsivity and hypersexuality (mania).

We cannot infer temporal relations between symptoms as our symptom data derive from one timepoint only. Some nodes measure overlapping constructs such an insomnia and sleep disturbances; these variables were however not strongly correlated, suggesting that clinicians are using these terms to report separate, but related, phenomena. As we limited our study to people presenting with unipolar depression, we were unable to assess the degree to which a transdiagnostic approach using NLP-derived symptom data could be better or worse at predicting future clinical outcomes compared with assessing initial ICD-10 diagnoses (TRANSD criterion 'S').[12] Our data derive

from a secondary mental healthcare setting and cannot be generalised to cases of unipolar depression that are managed exclusively within primary care. Future external validation studies should be performed as recommended by the TRANSD reporting guidelines (TRANSD criterion 'D').[17]

This study is also limited by the provenance of our data which restricts access to important clinical covariates. As our data are derived from a secondary mental healthcare case register, we do not have access to primary healthcare records. We, therefore, are unable to include comprehensive data on medications for people treated in primary care prior to referral to secondary care. Medication data are an important covariate because they can cause certain side effects such as agitation, and can increase risk of mania in some patients.[30 34 35]

Finally, we were also unable to extract data on substance use, symptoms of anxiety and the setting in which patients received their diagnosis (inpatient vs community). Substance use is a predictor of bipolar disorder,[36] while 'mixed anxiety and depressive disorder' is a common unipolar depression diagnosis, so they are important covariates to be included in future studies. Diagnosis setting may be a significant covariate when predicting rates of hospitalisation; patients who receive their diagnosis in inpatient settings may be more unwell and therefore hospitalised at greater rates. The majority of our sample is likely to have received their diagnosis in primary care settings.

## CONCLUSIONS

Our findings suggest that clinicians should screen for mania and diagnostically overlapping symptoms in patients who present with unipolar depression as these could be associated with increased risk of subsequently developing a psychotic or bipolar disorder and be associated with increased likelihood of psychiatric hospital admission.[12] The DSM-5 criteria for depression with mixed features does not include the symptoms 'agitation', 'irritability' or 'mood lability'.[2] Our study suggests patients diagnosed with MDD who present with these symptoms which 'overlap' both clinical depression and mania diagnoses may be at greater risk of developing a psychotic or bipolar disorder or at greater risk of psychiatric hospital admission. Such patients may require enhanced support and more tailored interventions to reduce risk of psychiatric admission. Symptom-level approaches to defining clinical phenotype may enable a better understanding of pathophysiology at the individual patient level, facilitate a more personalised treatment approach and better predict subsequent clinical outcomes.[37]

Future studies could evaluate whether the presence of mixed features, specifically mood instability and agitation, predict risk of onset of mania when treated with antidepressants. It is important to identify early warning signs for bipolar disorder as treatment approaches for bipolar disorder and unipolar depression differ considerably, and antidepressants can increase risk of mania.[30 34 35]

**Author affiliations**
[1]Department of Psychosis Studies, King's College London Institute of Psychiatry Psychology and Neuroscience, London, UK
[2]NIHR Maudsley Biomedical Research Centre, South London and Maudsley NHS Foundation Trust, London, UK
[3]Department of Psychiatry, University of Oxford, Oxford, UK
[4]Department of Psychological Medicine, King's College London Institute of Psychiatry Psychology and Neuroscience, London, UK

**Contributors** The study was conceived by RP. Data extraction and statistical analysis were performed by JI supervised by RP. Reporting of findings were carried out by RP, JI and AB. All authors (RP, JI, AB, MT, HS, MP, RS, PF-P and PM) contributed to study design, manuscript preparation and approved the final version. RP is guarantor.

**Funding** HS, MP, RS and PM have received funding from the National Institute for Health Research (NIHR) Biomedical Research Centre at South London and Maudsley NHS Foundation Trust and King's College London, which also supports the development and maintenance of the BRC Case Register RP has received funding from an NIHR Advanced Fellowship (NIHR301690), a Medical Research Council (MRC) Health Data Research UK Fellowship (MR/S003118/1) and a Starter Grant for Clinical Lecturers (SGL015/1020) supported by the Academy of Medical Sciences, The Wellcome Trust, MRC, British Heart Foundation, Arthritis Research UK, the Royal College of Physicians and Diabetes UK. RS has received funding from a Medical Research Council (MRC) Mental Health Data Pathfinder Award to King's College London, an NIHR Senior Investigator Award and the National Institute for Health Research (NIHR) Applied Research Collaboration South London (NIHR ARC South London) at King's College Hospital NHS Foundation Trust.

**Disclaimer** The views expressed are those of the authors and not necessarily those of the funders. The funders had no role in the design and conduct of the study; collection, management, analysis, and interpretation of the data; preparation, review, or approval of the manuscript; and decision to submit the manuscript for publication.

**Competing interests** All authors have completed the ICMJE uniform disclosure form at www.icmje.org/coi_disclosure.pdf and declare: RS has received funding from Janssen, GSK and Takeda outside the submitted work. RP has received funding from Janssen, Induction Healthcare and Holmusk outside the submitted work.

**Patient and public involvement** Patients and/or the public were involved in the design, or conduct, or reporting, or dissemination plans of this research. Refer to the Methods section for further details.

**Patient consent for publication** Not applicable.

**Ethics approval** This study involves human participants and was approved by Oxfordshire REC C (Ref: 18/SC/0372). The CRIS data resource received ethical approval as a deidentified dataset for secondary mental health research analyses from Oxfordshire REC C (Ref: 18/SC/0372). The study was approved by the SLaM BRC CRIS oversight committee. Consent is not required to analyse the deidentified dataset for approved research studies. Patients may opt-out of inclusion in the deidentified dataset.

**Provenance and peer review** Not commissioned; externally peer reviewed.

**Data availability statement** Data are available on reasonable request. Data may be obtained from a third party and are not publicly available. The data accessed by CRIS remain within an NHS firewall and governance is provided by a patient-led oversight committee. Access to data is restricted to honorary or substantive employees of the South London and Maudsley NHS Foundation Trust and governed by a local oversight committee who review and approve applications to extract and analyse data for research. Subject to these conditions, data access is encouraged and those interested should contact RS (robert.stewart@kcl.ac.uk), CRIS academic lead.

**ORCID iDs**
Rashmi Patel http://orcid.org/0000-0002-9259-8788
Jessica Irving http://orcid.org/0000-0002-2847-6508

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
