## [Reviewer comments · BMJ Open]

ARTICLE DETAILS

TITLE (PROVISIONAL)	Associations of presenting symptoms and subsequent adverse clinical outcomes in people with unipolar depression: a prospective natural language processing, transdiagnostic, network analysis of electronic health record data
AUTHORS	Patel, Rashmi; Irving, Jessica; Brinn, Aimee; Taylor, Matthew; Shetty, Hitesh; Pritchard, Megan; Stewart, Robert; Fusar-Poli, Paolo; McGuire, Philip

VERSION 1 – REVIEW

REVIEWER	Zhonggang Wang Shandong Daizhuang Hospital
REVIEW RETURNED	12-Nov-2021

GENERAL COMMENTS	1. was statistician involved in this paper? If so, please label. 2. Do you take care of patients' privacy ?
--

REVIEWER	Victoria Rodriguez King's College of London
REVIEW RETURNED	28-Nov-2021

GENERAL COMMENTS	The manuscript covers an interesting topic with some attractive results based on a novel way to make use of a natural language processing based clinical dataset. The study includes a sound sample representing a real clinical scenario with impressive size and a mean follow-up of over 8 years. Its main results on how unipolar depression with mixed traits could imply worse clinical outcomes have translational potential if replicated, and could lead to further research in identifying specific pathophysiology or target interventions for this subgroup. Major concerns: 1) The study intends to analyse how different existing symptoms in patients with unipolar depression determine subsequent bipolar or psychotic disorder or hospital admission. I got the impression from the introduction that the network analysis was suggested to identify those potentially more relevant symptoms before carrying further regressions for the clinical outcomes. However, the main results are presented using the dichotomised predictor variables of mania/overlapping/depression symptoms, which leave the output of the network analyses somehow unused or not relevant for the main aim of the study. The use of network analysis is well presented in the introduction, but authors should consider to explain how having employed the network analysis have contributed to the main results in the discussion.
--

2) Since patients were included after receiving their primary diagnosis of depression either in the community or inpatient service, I feel this is an important information that could have been included as covariate when analysing both clinical outcomes; since a previous hospital admission may be a factor predisposing for a further hospital admission or to manifest with BD or psychotic disorder in follow up.

3) When considering prediction of symptoms to develop Bipolar or Psychotic disorder in a clinical population already diagnosed with any form of depressive disorder, one key factor to confound with is medication. Given the evidence of how some antidepressants can not only trigger switch to mania but also induce other individual symptoms analysed in the current study, such as irritability, it is surprising this was not considered when designing the study.

4) It is noted the multiple regressions conducted for each group of symptoms and demographics, alongside the multivariable Cox regressions with multiple predictors for each of the outcomes, but it seems there has not been correction for multi testing. This should be mentioned/justified, and if not implemented, should be included in limitations.

5) Discussion is in my opinion the weakest part of the article, as it sometimes fails to provide interpretability of findings. There is no discussion around main findings of how presence of mania symptoms predicts subsequent bipolar/psychotic disorder or more hospital admissions. Additionally, it would be appreciated if the authors would provide more interpretation of findings of network analyses (pag 13, paragraph starting in line 20); or in the following. For instance, when the authors state: "Agitation and mood instability were the two overlapping symptoms that showed the highest betweenness centrality. However, betweenness and closeness show low stability in cross sectional data, wide confidence intervals, and inconsistency across datasets", what would be the take-home message for the reader?

6) Limitations don't cover important points such as lack of important clinical covariates in determining clinical outcomes such as medication, substance use or accounting for whether the initial contact was from community or inpatient services. Here should also be mentioned lack of multi testing correction as suggested above.

Minor comments:

7) The title missed one of the main outcomes (hospital admissions).

8) In the abstract, the main outcomes are presented in opposite order than in the rest of the text, consider inverting.

9) "Mixed anxiety and depressive disorder" is the third most frequent ICD-10 unipolar depression diagnosis, with up to 2,137 subjects. Given the relevance of this subgroup of patients, including anxiety (similarly as why mood instability was included) would have added very valuable information from the translational point of view. I do not intend the author to add this now, which would imply to record it from scratch and re-run analyses, but I would like to know if the authors considered to include anxiety as an additional symptom to explore, the reason why this was not included, and consider to place this as a limitation.

10) Since the TRANSD criterion is heavily mentioned in the study, it would be appreciated this to be included in supplementary material.

11) In Methods, page 7; part of the inclusion criteria is to not include those patients who develop bipolar disorder, mania or psychotic disorder within 3 months following diagnosis, but no clear rationale was given as per how it was established this threshold of 3 months

	vs 1 or 2 months. 12) In Methods, page 7-8; it may be me, but I can't understand the difference of precision and recall from how it is explained in the text; it comes to me as the same thing. Please revise and consider rephrasing to present it more clearly. 13) In Methods, page 9, the authors chose to dichotomous predictors as symptom groups had unequal numbers of constituent symptoms (therefore treating these predictors as continuous would have weighted one symptom group over the other). I don't know if the authors considered this, and I don't intend them to implement in the revised manuscript, but just as a suggestion, I wonder if a system of presenting proportion of fulfilled symptoms (from 0 to 1) within the categories of depressive, mania and overlapping would have added extra information; for instance two symptoms in mania or overlapping symptoms would be 0.4 whereas for depressive symptoms would be 0.2. 14) In Results, page 11. When presenting results of multivariable Cox hazard analyses, it is unclear the way it is presented. Is there a reason why having at least one symptom of mania, of overlapping symptoms or depression increased risk for either bipolar or psychotic disorder are not presented in the same sentence? (as it is presented for Cox hazard analysis for psychotic disorder as output) Do you consider these results not equally significant? If depression doesn't reach the corrected significant threshold this should be stated in here. 15) In Discussion: the most relevant results from the study seem to be that the presence of mixed traits (presence of mania and to a lesser extent of overlapping symptoms) are associated with worse clinical outcomes. However, results also show that symptoms of depression carry 1.31 times chances for subsequent psychotic or bipolar; and 1.41 times of hospital admission but not of compulsory. If this is not mentioned when summarising main results in discussion due to not considering that the p value reaches a new corrected significant threshold, this should be stated; otherwise it comes across as a bit misleading (having at least one symptom of depression is only mentioned for not increasing risk of compulsory admission, but not to state that it does increase risk of admission and of bipolar or psychotic disorder). Besides, it may have been more informative if symptoms of depression could have been used as a reference category, or having presented this as secondary analyses to get a better understanding of what the additional presence of mania or overlapping carries. 16) In Discussion, page 13, lines 43-44, authors wrote: "at least three of seven symptoms of mania including elation, grandiosity, pressured speech, flight of ideas, risk-taking behaviour and decreased need for sleep"; only six are mentioned, the seventh is missing. 17) In Discussion, when mentioning the results of black and asian ethnicity being almost three/two times more likely to be compulsory admitted, I missed a reflection over the fact that presence of mania, despite also predicts higher compulsory admission, was associated with being male and white; which here seem to imply that this is not due to black/asian population to present with more mania/overlapping symptoms, but other factors are taking place to influence those decision that require further investigation. 18) In Discussion, in the strengths subsection authors stated that: "Here, we identified 4,888 patients diagnosed with unipolar depression with symptoms of mania reported in their EHRs. Grouping these patients together with all patients with unipolar depression would have inadequately captured the heterogeneity of
--	---

	the unipolar depression diagnostic construct". This leads to conclude that patients with unipolar depression and symptoms of mania were not included, which is not the case; please consider rephrasing. 19) In Discussion, Strength subsection, pages 13 and 15 (page 14 is missing); line 54 from page 13 to line 11 in page 15 provide too unnecessary detail about a different study which doesn't contribute to the present study. I would suggest dropping this part and just mention the reference in relation to the specific strength applicable to the present study. 20) In Discussion, last paragraph of strength subsection (page 15, line 12-17); this doesn't fit in strengths; but could be placed instead in limitations when acknowledging the lack of information about medication as an important covariate (see comment above). 21) In Conclusions, I don't follow the point of the following sentence: "The DSM-5 criteria for depression with mixed features exclude patients at risk of developing a psychotic or bipolar disorder and who may need enhanced support and more tailored interventions to reduce risk of psychiatric hospital admission." 22) Figure 1: It is explained that edge thickness represents correlation magnitude, but I wanted to clarify it this is somewhat also based on results presented in Table 2, that seem to show the strongest edges for each symptom. That being the case, since as per Table 2, direction of analysis changes the strength of association (i.e hopelessness as node 1 and helplessness as node 2 is 1.65 whereas helplessness node 1 and worthlessness node 2 is 0.53), I wonder if the strongest association was chosen to define edge thickness in the figure. 23) Also related to Fig 1, it would help to get a more direct sense of the results to clarify the colours maybe by adding a circle with the colour next to the legend. 24) Tables: it is unclear what the three asterisk (***) mean, if it corresponds to a specific threshold, and this should be added to the legend of the tables. 25) Table 4: p value of being Asian should not be significant based on the CI95% - revise - .
--	---

VERSION 1 – AUTHOR RESPONSE

bmjopen-2021-056541

Reviewer: 1

Dr. Zhonggang Wang, Shandong Daizhuang Hospital

Comments to the Author:

1. Was statistician involved in this paper? If so, please label.

/*Thank you for your comments on our manuscript. Yes - data extraction and statistical analysis were performed by JI, a data scientist who has expertise in the network analysis methods used in this paper. We have described this in the "Author contributions" section of the manuscript:

"Data extraction and statistical analysis were performed by JI supervised by RP."*/

2. Do you take care of patients' privacy?

/*Yes - data for this study were derived from de-identified electronic health records in the South London and Maudsley (SLaM) NHS Foundation Trust Biomedical Research Centre (BRC) Case Register following approval by the SLaM BRC CRIS oversight committee. Patients may opt out of the de-identified dataset. This information can be found in the "Ethics approval" section of the paper:

"The CRIS data resource received ethical approval as a de-identified dataset for secondary mental health research analyses from Oxfordshire REC C (Ref: 18/SC/0372). The study was approved by the SLaM BRC CRIS oversight committee. Consent is not required to analyse the de-identified dataset for approved research studies. Patients may opt-out of inclusion in the de-identified dataset."*/

Reviewer: 2

Dr. Victoria Rodriguez, King's College of London

Comments to the Author:

The manuscript covers an interesting topic with some attractive results based on a novel way to make use of a natural language processing based clinical dataset. The study includes a sound sample representing a real clinical scenario with impressive size and a mean follow-up of over 8 years. Its main results on how unipolar depression with mixed traits could imply worse clinical outcomes have translational potential if replicated, and could lead to further research in identifying specific pathophysiology or target interventions for this subgroup.

/*Thank you for the above positive feedback and for your detailed and helpful suggestions on how we might improve the manuscript.*/*

Comments

1. The study intends to analyse how different existing symptoms in patients with unipolar depression determine subsequent bipolar or psychotic disorder or hospital admission. I got the impression from the introduction that the network analysis was suggested to identify those potentially more relevant symptoms before carrying further regressions for the clinical outcomes. However, the main results are presented using the dichotomised predictor variables of mania/overlapping/depression symptoms, which leave the output of the network analyses somehow unused or not relevant for the main aim of the study. The use of network analysis is well presented in the introduction, but authors should consider to explain how having employed the network analysis have contributed to the main results in the discussion.

/*We have restructured the discussion using subheadings to improve the clarity of the inferences related to the findings of the study. This should also help to address the points you have raised

subsequently. We have created a separate subheading entitled, “The relationship between symptoms of mania and depression” to discuss the findings related to the network analysis.*/

2. Since patients were included after receiving their primary diagnosis of depression either in the community or inpatient service, I feel this is an important information that could have been included as covariate when analysing both clinical outcomes; since a previous hospital admission may be a factor predisposing for a further hospital admission or to manifest with BD or psychotic disorder in follow up.

/*Thank you for this suggestion. Most individuals will have been diagnosed in primary care before being referred to the secondary care setting from which we have drawn our data. It is possible that patients who first presented to secondary mental health services as an inpatient will have been previously diagnosed with depression in primary care. We did not have access to primary care data to evaluate whether patients had been previously diagnosed with depression prior to presenting to secondary mental healthcare services. We have acknowledged this as a limitation of the study:

“This study is also limited by the provenance of our data which restricts access to important clinical covariates. As our data is derived from a secondary mental healthcare case register, we do not have access to primary healthcare records.”*/

3. When considering prediction of symptoms to develop Bipolar or Psychotic disorder in a clinical population already diagnosed with any form of depressive disorder, one key factor to confound with is medication. Given the evidence of how some antidepressants can not only trigger switch to mania but also induce other individual symptoms analysed in the current study, such as irritability, it is surprising this was not considered when designing the study.

/*Thank you for highlighting this important point. Accurate medication data is difficult to establish as patients are often treated with antidepressant therapy in primary care settings. We did not have access to primary care data in our study and so were unable to provide comprehensive data on medications for people treated in primary care prior to referral to secondary care. We have now included this explanation in the limitations section of our discussion (please see response to comment 2, above).*/

4. It is noted the multiple regressions conducted for each group of symptoms and demographics, alongside the multivariable Cox regressions with multiple predictors for each of the outcomes, but it seems there has not been correction for multi testing. This should be mentioned/justified, and if not implemented, should be included in limitations.

/*Thank you for highlighting this. We had corrected p values for multiple testing using the Bonferroni method in the original analysis. We have updated the methods section to describe this:

“Multivariable analyses were corrected for multiple-testing using the Bonferroni method.”*/

5. Discussion is in my opinion the weakest part of the article, as it sometimes fails to provide interpretability of findings. There is no discussion around main findings of how presence of mania symptoms predicts subsequent bipolar/psychotic disorder or more hospital admissions. Additionally, it would be appreciated if the authors would provide more interpretation of findings of network analyses (pag 13, paragraph starting in line 20); or in the following. For instance, when the authors state: “Agitation and mood instability were the two overlapping symptoms that showed the highest betweenness centrality. However, betweenness and closeness show low stability in cross sectional data, wide confidence intervals, and inconsistency across datasets”, what would be the take-home message for the reader?

/*We have restructured the discussion section using subheadings to improve clarity: “Association of symptoms with psychotic or bipolar disorder diagnosis” and “Association of symptoms with subsequent hospital admissions”. To be clear, we cannot conclude that the presence of symptoms are predictive of subsequent outcomes based on the real-world data we analysed and we have worded our interpretation based on the associations of symptoms with outcomes. Under the heading “The relationship between symptoms of mania and depression”, we have included a ‘take-home message’ as suggested. The take-home message is as follows:

“This finding may suggest that clinicians should be especially vigilant in identifying symptoms of agitation or mood instability in patients with unipolar depression, as they could be indicators associated with of subsequent transition to diagnosis of bipolar disorder and/or increased risk of subsequent psychiatric hospitalisation.”*/

6. Limitations don’t cover important points such as lack of important clinical covariates in determining clinical outcomes such as medication, substance use or accounting for whether the initial contact was from community or inpatient services. Here should also be mentioned lack of multi testing correction as suggested above.

/*We have expanded our limitations section to include the above suggestions. We had corrected for multi-testing in our analyses, and this has now been highlighted in the methods section.

“This study is also limited by the provenance of our data which restricts access to important clinical covariates. As our data is derived from a secondary mental healthcare case register, we do not have access to primary healthcare records. We therefore are unable to include comprehensive data on medications for people treated in primary care prior to referral to secondary care. Medication data is an important covariate because they can cause certain side effects such as agitation, and can increase risk of mania in some patients. [22,27,28]

Finally, we were also unable to extract data on substance use, symptoms of anxiety, and the setting in which patients received their diagnosis (inpatient vs community). Substance use is a predictor of

bipolar disorder [30], whilst 'mixed anxiety and depressive disorder' is a common unipolar depression diagnosis, so they are important covariates to be included in future studies. Diagnosis setting may be a significant covariate when predicting rates of hospitalisation; patients who receive their diagnosis in inpatient settings may be more unwell and therefore hospitalised at greater rates. The majority of our sample is likely to have received their diagnosis in primary care settings."*/

7. The title missed one of the main outcomes (hospital admissions).

/*We have modified the title to refer to 'subsequent adverse clinical outcomes', which refers both to diagnosis of bipolar or psychotic disorder and/or psychiatric hospital admission.*/*

8. In the abstract, the main outcomes are presented in opposite order than in the rest of the text, consider inverting.

/*Thank you for highlighting this inconsistency. We have inverted the order to match the rest of the text.*/*

9. "Mixed anxiety and depressive disorder" is the third most frequent ICD-10 unipolar depression diagnosis, with up to 2,137 subjects. Given the relevance of this subgroup of patients, including anxiety (similarly as why mood instability was included) would have added very valuable information from the translational point of view. I do not intend the author to add this now, which would imply to record it from scratch and re-run analyses, but I would like to know if the authors considered to include anxiety as an additional symptom to explore, the reason why this was not included, and consider to place this as a limitation.

/*We agree that anxiety is an important symptom to capture. We have included this as a limitation to our study and suggest that future studies aim to include this symptom in their analyses:

"Finally, we were also unable to extract data on substance use, symptoms of anxiety, and the setting in which patients received their diagnosis (inpatient vs community). Substance use is a predictor of bipolar disorder [30], whilst 'mixed anxiety and depressive disorder' is a common unipolar depression diagnosis, so they are important covariates to be included in future studies."*/

10. Since the TRANSD criterion is heavily mentioned in the study, it would be appreciated this to be included in supplementary material.

/*Thank you for this helpful suggestion. We have now included the TRANSD criterion in the supplementary material.*/*

11. In Methods, page 7; part of the inclusion criteria is to not include those patients who develop bipolar disorder, mania or psychotic disorder within 3 months following diagnosis, but no clear rationale was given as per how it was established this threshold of 3 months vs 1 or 2 months.

/*Symptom data were drawn from a period within three months either side of the index date, as symptoms presenting in this time are more likely to inform the initial diagnosis. Clinical assessment can take a few months and while there may be evidence of bipolar disorder at initial assessment, a formal diagnosis may not be documented at the first clinical assessment. Follow-up data were obtained from three months following the index date to ensure predictor symptom were temporally separate from the outcome data:

“Symptom data were drawn from a period within three months either side of the index date, as symptoms presenting in this time are more likely to inform initial presentation.”

“Follow-up data were obtained from 3 months following the index date for each patient up to 31st March 2021 to ensure predictor symptom data were temporally separate from outcome data.”*/

12. In Methods, page 7-8; it may be me, but I can't understand the difference of precision and recall from how it is explained in the text; it comes to me as the same thing. Please revise and consider rephrasing to present it more clearly.

/*We have rephrased this explanation to highlight the difference between precision and recall more clearly:

“Their performance is measured through precision and recall. In this case, precision refers to the number of true positive instances that the NLP application has identified, divided by the total number of instances retrieved by the NLP application (both true positive and false positive instances). Recall refers to the number of true positive instances identified by the NLP application, divided by the total number of existing true positives in the entire sample. Precision and recall statistics for the NLP applications employed in this as well as further details on their derivation, are provided in the CRIS Natural Language Processing Applications Library.[16]”*/

13. In Methods, page 9, the authors chose to dichotomous predictors as symptom groups had unequal numbers of constituent symptoms (therefore treating these predictors as continuous would have weighted one symptom group over the other). I don't know if the authors considered this, and I don't intend them to implement in the revised manuscript, but just as a suggestion, I wonder if a system of presenting proportion of fulfilled symptoms (from 0 to 1) within the categories of depressive, mania and overlapping would have added extra information; for instance two symptoms in mania or overlapping symptoms would be 0.4 whereas for depressive symptoms would be 0.2.

/*We choose the exposure to be the presence of one or more symptoms rather than considering proportions of symptoms within each symptom group. Our defined symptom groups are not validated for comparison as symptom scales. Moreover, as symptom groups are different sizes, a proportion of symptoms in one symptom group may not relate to the same proportion in another symptom group.*/

14. In Results, page 11. When presenting results of multivariable Cox hazard analyses, it is unclear the way it is presented. Is there a reason why having at least one symptom of mania, of overlapping symptoms or depression increased risk for either bipolar or psychotic disorder are not presented in the same sentence? (as it is presented for Cox hazard analysis for psychotic disorder as output) Do you consider these results not equally significant? If depression doesn't reach the corrected significant threshold this should be stated in here.

/*We have modified our presentation of these results to avoid confusion:

“In decreasing order of risk, individuals with at least one symptom of mania, at least one overlapping symptom, or at least one symptom of depression had significantly increased risk for bipolar or psychotic disorder.”*/

15. In Discussion: the most relevant results from the study seem to be that the presence of mixed traits (presence of mania and to a lesser extent of overlapping symptoms) are associated with worse clinical outcomes. However, results also show that symptoms of depression carry 1.31 times chances for subsequent psychotic or bipolar; and 1.41 times of hospital admission but not of compulsory. If this is not mentioned when summarising main results in discussion due to not considering that the p value reaches a new corrected significant threshold, this should be stated; otherwise it comes across as a bit misleading (having at least one symptom of depression is only mentioned for not increasing risk of compulsory admission, but not to state that it does increase risk of admission and of bipolar or psychotic disorder). Besides, it may have been more informative if symptoms of depression could have been used as a reference category, or having presented this as secondary analyses to get a better understanding of what the additional presence of mania or overlapping carries.

/*Thank you for highlighting this. We have now included all depression findings in the discussion. We agree that it would be an interesting and worthwhile analysis to use symptoms of depression as a reference category. However, we chose this method because we wanted to understand the associations between presence and absence of our particular symptom categories, rather than presence compared to depression:

“Overlapping symptoms and depression symptoms were associated with increased risk of subsequent psychiatric hospital admissions. However, symptoms of depression were not associated with increased risk of compulsory psychiatric hospital admission.”*/

16. In Discussion, page 13, lines 43-44, authors wrote: “at least three of seven symptoms of mania including elation, grandiosity, pressured speech, flight of ideas, risk-taking behaviour and decreased need for sleep”; only six are mentioned, the seventh is missing.

/*Many thanks. We have now included the seventh symptom in the text:

“the presence of at least three of seven symptoms of mania including elevated mood, grandiosity, pressured speech, flight of ideas, increased energy, risk-taking behaviour and decreased need for sleep.”*/

17. In Discussion, when mentioning the results of black and asian ethnicity being almost three/two times more likely to be compulsory admitted, I missed a reflection over the fact that presence of mania, despite also predicts higher compulsory admission, was associated with being male and white; which here seem to imply that this is not due to black/asian population to present with more mania/overlapping symptoms, but other factors are taking place to influence those decision that require further investigation.

/*The risk for compulsory hospital admission is based on multiple variables. We have measured some of these variables in our study e.g., ethnicity, gender. However, there are other variables which we have not been able to measure, such as medication and social factors. After accounting for ethnicity in the multivariable analysis, mania is still associated with compulsory admission. We agree that this finding requires further investigation.*/*

18. In Discussion, in the strengths subsection authors stated that: “Here, we identified 4,888 patients diagnosed with unipolar depression with symptoms of mania reported in their EHRs. Grouping these patients together with all patients with unipolar depression would have inadequately captured the heterogeneity of the unipolar depression diagnostic construct”. This leads to conclude that patients with unipolar depression and symptoms of mania were not included, which is not the case; please consider rephrasing.

/*We have rephrased this sentence to provide better clarity:

“Here, we identified 4,888 patients diagnosed with unipolar depression with symptoms of mania reported in their EHRs. We were able to stratify these patients into a separate group and represent the heterogeneity of unipolar depression. As a result, we were able to identify differences in clinical outcomes between patients diagnosed with the same heterogeneous diagnostic construct.”*/

19. In Discussion, Strength subsection, pages 13 and 15 (page 14 is missing); line 54 from page 13 to line 11 in page 15 provide too unnecessary detail about a different study which doesn't contribute to the present study. I would suggest dropping this part and just mention the reference in relation to the specific strength applicable to the present study.

/*We agree that there was unnecessary detail in this paragraph. We have removed these details to improve clarity.*/

20. In Discussion, last paragraph of strength subsection (page 15, line 12-17); this doesn't fit in strengths; but could be placed instead in limitations when acknowledging the lack of information about medication as an important covariate (see comment above).

/*We agree that this paragraph was incorrectly placed in strengths. We have removed this paragraph and included information about medication as an important covariate in the limitations subsection.*/

21. In Conclusions, I don't follow the point of the following sentence: "The DSM-5 criteria for depression with mixed features exclude patients at risk of developing a psychotic or bipolar disorder and who may need enhanced support and more tailored interventions to reduce risk of psychiatric hospital admission."

/*We agree that the wording of this sentence was unclear. The intended point was that the DSM-5 criteria does not include key overlapping symptoms such as agitation or irritability. For this reason, it may not capture a subset of patients with these symptoms who are at increased risk of transition to bipolar/psychiatric hospital admission. We have modified the statement to make this clearer:

"The DSM-5 criteria for depression with mixed features does not include the symptoms 'agitation', 'irritability', or 'mood lability'. [2] Our study suggests patients diagnosed with MDD who present with these symptoms which 'overlap' both clinical depression and mania diagnoses may be at greater risk of developing a psychotic or bipolar disorder or at greater risk of psychiatric hospital admission. Such patients may require enhanced support and more tailored interventions to reduce risk of psychiatric admission."*/

22. Figure 1: It is explained that edge thickness represents correlation magnitude, but I wanted to clarify it this is somewhat also based on results presented in Table 2, that seem to show the strongest edges for each symptom. That being the case, since as per Table 2, direction of analysis changes the strength of association (i.e hopelessness as node 1 and helplessness as node 2 is 1.65 whereas helplessness node 1 and worthlessness node 2 is 0.53), I wonder if the strongest association was chosen to define edge thickness in the figure.

/*Table 2 provides estimates of the strength of association between nodes and provides extra information about how these estimates change with directionality. Figure 1 contains weighted lines which represent the strength of the relationship between nodes. However, in figure 1 the edges are non-directional. The nodes are connected by an edge to represent their mutual relationship and edges are thicker where this relationship is stronger.*/

23. Also related to Fig 1, it would help to get a more direct sense of the results to clarify the colours maybe by adding a circle with the colour next to the legend.

/*Thank you, we have already provided colour coding in the circles of legend of Figure 1.*/

24. Tables: it is unclear what the three asterisk (***) mean, if it corresponds to a specific threshold, and this should be added to the legend of the tables.

/*Thanks for highlighting this. We have now included a legend in the tables to define what the asterisks mean.*/

25. Table 4: p value of being Asian should not be significant based on the CI95% - revise

/*We have revised this – thank you.*/